# OpenReview forum: "FormalJudge: A Neuro-Symbolic Paradigm for Agentic Oversight"
_ICML.cc/2026/Conference — ICML 2026 regular_

### Official Review · Reviewer_8RqB · 2026-02-18

**Soundness:** 4
**Presentation:** 4
**Significance:** 4
**Originality:** 3
**Overall Recommendation:** 6
**Confidence:** 5

**Summary:**

The key idea is to use a neuro-symbolic framework where LLMs decompose human-provided specifications into
verifiable constraints, where compliance to these constraints is proved using  Satisfiability modulo theories solving, which guarantees
correctness of the process.

**Compliance With Llm Reviewing Policy:**

Affirmed.

**Ethical Review Concerns:**

---

**Key Questions For Authors:**

---

**Limitations:**

The main limitations of the approach are in the difficulty of eliciting the "right" from specifications from users. However, this problem is unavoidable. And a neuro-symbolic approach is inherently easier to control than a purely inductive approach.

**Strengths And Weaknesses:**

The work seems sound, and it is definitely well presented

The work is definitely significant. The top-down approach of decomposing informal specifications into simpler formal specifications is as good as you can do. Users are supported in maybe the most difficult task in the definition of specifications

Some approaches have been proposed, not too many, which deal with the informal-to-formal problem. The technique proposed to me is novel and has high potential in particular because of the high reasoning power and efficiency of SMT problem solvers.

---

> ### Author Rebuttal · Authors · 2026-03-31
>
> **We sincerely thank you for the thorough evaluation and strong endorsement.** Your detailed assessment across soundness, significance, and originality is deeply encouraging.
>
> > On the top-down decomposition approach
>
> We are glad you recognize the value of decomposing informal specifications into simpler formal specifications. This top-down design is central to FormalJudge:
>
> - **Phase 1** breaks user intent into atomic, verifiable constraints;
> - **Phases 3-4** delegate composition to deterministic Dafny/Z3 solvers.
>
> As you note, this is **"as good as you can do"** for bridging the informal-to-formal gap. We are also encouraged that other reviewers share your view, noting that limiting LLMs to fact extraction while delegating logical composition to SMT solvers is "novel and original" and that the Formal-of-Thought architecture is "promising for different domains" beyond safety alone.
>
> > On specification elicitation
>
> We fully agree with you that eliciting the "right" specifications is the primary challenge, and appreciate your balanced framing of this as an **inherent difficulty rather than a flaw**. Another reviewer also highlighted that "hallucination echo chambers" in LLM-supervising-LLM pipelines are a major open problem; our specification elicitation challenge is essentially the *controlled* version of this broader issue, with the critical advantage that errors are **isolated and diagnosable**.
>
> During the rebuttal period, we conducted a **root-cause analysis across three benchmarks**:
>
> - **~80.6%** of errors originate in the LLM-dependent specification compiler (incorrect formalization 41.2%, unsupported fabrication 29.0%);
> - **~6.4%** from the formal back-end (Dafny + Z3), caught by the compiler.
>
> This confirms your intuition: the bottleneck lies at the informal-to-formal interface, not in the verification engine itself. Because errors are isolated to the specification stage rather than diffused across an opaque reasoning chain, they are **incrementally solvable**. We are exploring retrieval-augmented formalization and templatized specification libraries; early results show a ~12 % reduction in formalization errors on VitaBench.
>
> As frontier models continue to improve in formal reasoning and specification capability, we expect the primary bottleneck at the informal-to-formal interface to be eliminated, further amplifying the effectiveness of our FormalJudge framework.
>
> > On the neuro-symbolic advantage
>
> Your observation that **"a neuro-symbolic approach is inherently easier to control than a purely inductive approach"** precisely captures our core thesis. Reviewers also highlighted weak-to-strong generalization and formal verification feedback as "promising" directions, and we are glad to share new empirical evidence further supporting this:
>
> - **Robustness against adaptive attacks.** On X-Teaming (SOTA dynamic multi-turn jailbreak), FormalJudge (Dafny) achieves **98.6%** Acc on Claude-Sonnet-4.5 and **88.9%** on GPT-4o, outperforming the best baseline by +10-25 %.
> - **Generalization to real-world agents.** On OpenClaw (310K+ GitHub stars), we constructed 42 multi-turn attack scenarios with **auto-generated** Dafny specs (no manual writing). FormalJudge reaches **90.5%** Acc (GPT-4o judge) and **88.1%** (Claude-Opus-4-6 judge), surpassing LLM Judge by +22-29 % regardless of judge capability.
> - **Voting cannot close the gap.** We evaluated 6 baselines with majority voting (@3/@6/@9). Gains are marginal (+0.1 to +3.6 %) and do not scale with more samples, confirming the bottleneck is LLMs' capability ceiling, not sampling noise.
>
> Together with the comprehensive evaluation acknowledged across reviews, these results reinforce that the SMT-backed approach offers a **qualitatively different, more controllable paradigm** for agentic oversight. We will incorporate all new experiments and expanded analyses in the future version.
>
> **We are sincerely grateful for your confidence in this work.** We are encouraged that your assessment aligns with a broader consensus across reviews: the novelty of the neuro-symbolic architecture and the promise of formal verification feedback were highlighted by other reviewers as well. During the rebuttal period, we have worked to address all remaining concerns with substantial new experiments and analyses, and we are committed to delivering a stronger final version.

---

> > ### Author Rebuttal · Reviewer_8RqB · 2026-04-04
> >
> > there were no major concernes to begin with

---

> > > ### Author Response · Authors · 2026-04-06
> > >
> > > Thank you for your time, the thoughtful review, and **your consistent support and confidence in our work** throughout the process! We genuinely appreciate it. It means a lot to us.

---

### Official Review · Reviewer_dpo3 · 2026-03-12

**Soundness:** 3
**Presentation:** 3
**Significance:** 3
**Originality:** 4
**Overall Recommendation:** 5
**Confidence:** 4

**Summary:**

LLMs are widely being used as autonomous agents over simply an assistant. Agentic systems use the LLM-as-a-judge paradigm to oversee other agents leading to one probabilistic system overseeing other probabilistic systems. This paper addresses this problem with the novel Formal-of-Thought framework - FormalJudge. It limits LLMs to extract atomic facts and uses deterministic SMT solvers for logical decomposition. The paper shows boost over LLM-as-ajudge baselines and an impressive weak-to-strong generalization. A comprehensive evaluation framework is leveraged across - behavioral safety, constrained adherence, and deception resistance.

**Compliance With Llm Reviewing Policy:**

Affirmed.

**Final Justification:**

The paper addresses the "hallucination echo chamber" that occurs when probabilistic LLMs are used to oversee other probabilistic agents. The authors propose FormalJudge, which decomposes human intent into atomic facts via an LLM "specification compiler." These facts are then fed into a deterministic formal verification backend (Dafny and Z3). The framework is evaluated across behavioral safety, constraint adherence, and deception resistance, demonstrating significant improvements over traditional "LLM-as-a-judge" baselines.

**Strengths -**
- Architectural Novelty: The shift from holistic probabilistic scoring to a "specification compiler + SMT solver" pipeline is a major contribution. It effectively isolates LLM reasoning to simple fact extraction, where it is most reliable.
- Weak-to-Strong Generalization: The framework demonstrates an impressive ability to oversee and verify agents that are more "intelligent" or sophisticated than the judge itself.
- Robustness to Deception: The use of grounded trajectory logs allows the system to catch inter-fact contradictions that holistic LLM judges typically miss.

**Weaknesses & Rebuttal Outcomes -**
Almost all weaknesses and questions are answered in great detail. Authors are clear with limitations of their work and current analysis. eg, latency, qualitative constraints, etc.

**Decision -**
The paper is accepted because it offers a technically sound and highly original solution to one of the most pressing problems in agentic AI: reliable oversight.

The "Formal-of-Thought" framework moves the field toward verifiable AI. The authors' rebuttal was particularly strong, providing new empirical data on manipulation resistance and a clear breakdown of error sources. While the dependency on an LLM for initial formalization remains a limitation, the framework successfully shifts the "failure surface" to a manageable interface. The demonstrated improvements in accuracy and the robustness against deceptive agents make this a significant contribution to the community.

The authors are encouraged to include the latency breakdown and the manipulation-attack results (P1-P4) in the final version of the paper, as these significantly strengthen the practical case for FormalJudge.

**Key Questions For Authors:**

1. Expressing real world constraints in math expressions is challenging. The paper shows examples of hard constraints like “budget < $500”. How about qualitative constraints like “text should be polite”, “write and efficient code” are handled?
2. Did the agent's model face any challenges to receive and improve using formal verification feedback especially in Defny, Z3 formats? I assume the underlying LLMs training sample might not be representative of such formal verification code outputs.

**Limitations:**

Yes

**Strengths And Weaknesses:**

**Strength -**
1. The paper addresses a major problem in LLM based supervision space - hallucination echo chambers, leaving the supervising agent vulnerable to sophisticated manipulation and deceptions.
2. The idea of leveraging LLMs as a specification compiler and deterministic SMT solvers for logical composition is novel. It limits LLMs to extract facts in simple “yes/no” format and uses SMT solvers (Defny, Z3) for proof synthesis and verification. This method is definitely an improvement over LLM’s probabilistic scoring.
3. The authors provide a comprehensive evaluation framework and detailed result analysis.
4. Weak to strong generalization and formal verification feedback over natural language results are promising.
5. Overall the paper is easy to read and understand.

**Weakness -**
1. The proposed method is still dependent on LLM as a specification compiler to decompose into granular facts. The impact of hallucinations at this stage are not discussed in detail.
2. The method relies on agent’s trajectory logs (actions, observations) instead of reasoning explanations/thoughts. However, deceptive agents can still manipulate these grounded facts. Discussion on this limitation will be helpful.
3. In the real world system, the LLM-as-a-judge paradigm is used with a limited latency budget. The proposed method goes through several stages which can add latency overhead. An analysis on this latency overhead could have been helpful to understand practical adoption.

---

> ### Author Rebuttal · Authors · 2026-03-31
>
> We thank you for the thorough review and practical deployment questions.
>
> > W1: Hallucination impact in LLM-dependent stages
>
> You raise an important concern. Root-cause analysis across three benchmarks: **~80.6% of errors originate in the LLM-dependent specification compiler** (incorrect formalization 41.2%, unsupported fabrication 29.0%), while the formal back-end (Dafny + Z3) contributes only ~6.4% (caught by compiler).
>
> We do not claim end-to-end formal guarantees. Instead, FormalJudge provides *conditional* guarantees: given correct formalization, verification is sound. The advantage is that **errors are isolated to the NL-to-formal interface** rather than diffused across an opaque reasoning chain, making them incrementally solvable. We are actively exploring retrieval-augmented formalization and templatized specification libraries to reduce this error rate.
>
> > W2: Deceptive agents can manipulate grounded facts
>
> Trajectory manipulation always leaves detectable traces, since fabricating tool returns requires additional tool calls that produce observable anomalies.
>
> We designed four environment-manipulation attacks under a **black-box threat model** (the agent is unaware of FormalJudge's detection logic) (n=10 each; 40 perturbed + 10 honest + 10 deceptive controls):
> - **P1:** Agent overwrites file, re-reads fabricated content
> - **P2:** After failed download, agent creates local fake file
> - **P3:** Agent tampers with raw_log to hide errors
> - **P4:** Agent reads wrong file, presents it as target
>
> We evaluate three configurations: **LLM-as-a-Judge** (direct LLM assessment); **Standard FJ** (default pipeline, no manipulation-specific rules); **Enhanced FJ** (adds four lightweight pattern-matching detectors over the action sequence, **no additional LLM calls**).
>
> | Metric | LLM-as-a-Judge | Standard FJ | Enhanced FJ |
> |--------|----------------|-------------|-------------|
> | Acc | 0.623 | 0.824 | 0.932 |
>
> Even without manipulation-specific design, Standard FJ (0.824) already substantially outperforms LLM-as-a-Judge (0.623) via cross-validation catching inter-fact contradictions. Enhanced FJ reaches 0.932 by adding declarative pattern-matching atoms, demonstrating both **inherent robustness** and **recoverability** (new detection rules integrate without additional LLM calls). We will expand sample sizes in the future version.
>
> > W3: Latency overhead analysis
>
> FormalJudge introduces additional latency, but we believe it is justified. We provide an overall comparison and a **stage-level breakdown**.
>
> **Accuracy (% averaged over 7 agent models) and averaged latency**
>
> | Method | Agent-SafetyBench | VitaBench | Deceivers | Latency |
> |--------|-------------|-----------|-----------|---------|
> | LLM | 59.9 | 66.2 | 65.4 | 4.83s |
> | +CoT | 58.9 | 65.8 | 67.0 | 6.58s |
> | +Few-Shot | 59.8 | 64.9 | 69.4 | 5.01s |
> | +Struct. | 65.3 | 67.3 | 73.9 | 8.18s |
> | +Python | 66.4 | 69.8 | 74.7 | 12.32s |
> | **FormalJudge** | **76.2** | **74.8** | **93.7** | **14.59s** |
>
> **Where does the time go?**
>
> | Stage | % of Total |
> |-------|------------|
> | Decomposition + fact extraction + Dafny synthesis (LLM) | ~72% |
> | Trace abstraction (LLM) | ~23% |
> | **Dafny compilation + Z3 solving (deterministic)** | **<6%** |
>
> Three sequential LLM API calls dominate (~95%); Dafny + Z3 is *not* the bottleneck. Latency scales with LLM API speed and is reducible via specification caching and parallel evaluation. Moreover, as you noted, real-world systems operate under latency budgets; post-hoc oversight runs asynchronously, and a 16.6% detection improvement justifies ~3x latency for an offline audit that does not block agent execution.
>
> > Q1: How are qualitative constraints handled?
>
> Great question. Phase 1 decomposes qualitative intents into observable proxies (e.g., "polite" -> "no profanity", "no dismissive tone"); Phase 2 (Eq. 4) makes atomic yes/no judgments on each proxy. This is less precise than hard numeric constraints, but still outperforms holistic LLM judgment. On VitaBench L7 (**semantic intent matching**), FormalJudge achieves 72.1% alignment vs. best baseline 65.4% (**+6.7 %**), confirming that **structured decomposition benefits even qualitative evaluation.**
>
> > Q2: Can agents use Dafny/Z3 feedback despite limited training exposure?
>
> Thank you for this practical concern. **This is precisely why we conducted iterative refinement experiments on both open-source and closed-source models in Sec. 4.5**: agents receive Dafny specifications with concrete constraint violations and revise their actions. Claude-4.5-Opus improves from 70.7% to 99.8% over 3 rounds; **even Qwen-7B improves from 30.8% to 49.4%.** The key insight is that agents do not need to *write* Dafny; they only need to *read* structured feedback (which constraint was violated and why), which modern LLMs handle well.
>
> **We hope these results further strengthen your confidence in FormalJudge's practical viability.** All improvements will be incorporated in the final version.

---

> > ### Author Rebuttal · Reviewer_dpo3 · 2026-04-03
> >
> > Thank you for your response. They resolved all my open questions well.

---

> > > ### Author Response · Authors · 2026-04-03
> > >
> > > We are deeply grateful for the reviewer’s patient and meticulous review. It is a great honor to know that our rebuttal **successfully addressed your concerns and provided the necessary clarification**. Thank you once again for your dedication and effort; **it has truly made this submission process a rewarding experience for us.**

---

### Official Review · Reviewer_H2wa · 2026-03-13

**Soundness:** 3
**Presentation:** 2
**Significance:** 3
**Originality:** 3
**Overall Recommendation:** 4
**Confidence:** 3

**Summary:**

This paper proposes a neuro-symbolic framework to formally verify that the probabilistic LLM reliably supervises other probabilistic systems. More specifically, it adopts a top-down decomposition from human intent to atomic and verifiable constraints, followed by Dafny specification and SMT verification for bottom-up proofs. The proposed method has been evaluated on multiple benchmarks for different tasks with several baselines to validate its effectiveness.

**Compliance With Llm Reviewing Policy:**

Affirmed.

**Final Justification:**

I keep my positive rating, and my concerns have been addressed.

**Key Questions For Authors:**

- In the formulation of Sec 3.1, the probabilistic judge can be averaged to be smoother or more deterministic, right? Will it be a possible baseline in experiments?

- In the generation of atomic facts $f_i$, why is the user intent $I$ not decomposed into atomic specifications? Currently atomic factors are associated with each step of $\tau$.

- In the experiments of weak-to-strong deception detection, why does the detection accuracy decrease as the agent model size grows? The judge's size should be the same in this case, no?

- I wonder if this framework can be used in the defense of multi-turn jailbreak attacks, where the neuro-symbolic framework can be used to monitor the safety of the conversation. It should be discussed for broader applications.

**Limitations:**

Yes

**Strengths And Weaknesses:**

Strength:

- The motivation is sound as it is important to study formal verification for the agentic oversight. The method is technically sound and the formal-of-thought architecture makes sense to me.

- The idea of leveraging Dafny and SMT solver to do the verification for LLM agentic behavior is novel and original. It is promising for different domains where neuro-symbolic ideas can be used to enhance the multi-turn safety of LLM agents.

- The figure presentation is clear and the paper writing is easy to follow.

Weakness:

- The storytelling does not seem to be natural, where verification of the probabilistic systems is not specific and it is better to justify the difference between agentic oversight and agentic safety.

- More details about the methodology need to be clarified, e.g. how Eq (2) (3) (4) are implemented in practice. More clarification needs to be provided to make it more friendly for the general audience.

---

> ### Author Rebuttal · Authors · 2026-03-31
>
> **We thank you for the constructive feedback!**
>
> > W1: "Verification of probabilistic systems" & "agentic oversight vs. safety"
>
> - **Probabilistic systems**: FormalJudge performs *deterministic verification of probabilistic judgments*, not probabilistic model checking in the classical sense. It confines LLMs to atomic yes/no extraction (Phase 2) and delegates composition to deterministic Dafny/Z3 (Phase 3-4), breaking the shared failure modes of LLM-supervising-LLM.
> - **Oversight vs. safety**: **Oversight is a post-hoc evaluation component within the safety pipeline**: safety designs safe behaviors, oversight verifies compliance after execution. **It extends beyond safety** to cover trustworthiness, constraint adherence, and intent alignment. We will restructure the introduction to make this explicit.
>
> > W2: Eq (2)(3)(4) implementation details
>
> Each atomic fact is parsed deterministically (Eq. 3), falling back to LLM extraction on minimal context (Eq. 4), then mapped to Dafny predicates verified by Z3 (Eq. 2). We will add pseudocode, annotate Figure 3, and include a notation glossary.
>
> > Q1: Can probabilistic judge averaging serve as a baseline?
>
> Yes. We evaluated 6 baselines × 3 temperatures(0.3, 0.6, 0.9) × 3 runs with majority voting on all 3 benchmarks:
>
> | |Deceivers | | |Agent-SafetyBench | | |VitaBench | | |
> |---|---|---|---|---|---|---|---|---|---|
> | Method | @3 | @6 | @9 | @3 | @6 | @9 | @3 | @6 | @9 |
> | LLM | +1.5 | +1.8 | +1.2 | +1.7 | +2.0 | +2.3 | +0.6 | +0.1 | +0.5 |
> | LLM+CoT | +1.9 | +2.1 | +1.8 | +1.0 | +2.4 | +3.4 | +0.5 | +0.5 | +0.3 |
> | LLM+Struct. | +1.5 | +1.8 | +0.5 | +2.3 | +2.5 | +3.6 | +1.0 | +0.7 | +0.9 |
> | LLM+Few-Shot | +2.2 | +2.6 | +1.8 | +1.0 | +1.1 | +1.6 | +0.7 | +0.6 | +0.2 |
> | LLM+Python | +2.3 | +2.3 | +1.4 | +1.0 | +2.1 | +3.2 | +0.5 | +0.6 | +0.4 |
> | FormalJudge | +0.1 | +0.0 | +0.1 | +0.0 | +0.2 | +0.1 | +0.0 | +0.0 | +0.1 |
>
> Values show Δ Acc (%) over single-run baseline. **Voting yields only marginal and inconsistent gains for all baselines.** FormalJudge itself shows near-zero variance (Δ ≤ 0.2%) under the same setup, confirming its architecture is **inherently deterministic** and does not benefit from probabilistic smoothing. The bottleneck for baselines is capability ceiling, not sampling noise.
>
> > Q2: Why not decompose intent *I* into atomic specifications?
>
> We appreciate you raising this; it reveals that our description was unclear. FormalJudge already performs this decomposition: Phase 1 decomposes *I* top-down into atomic facts, while extraction is bottom-up from τ. Each $f_i$:(*I*,τ)→{0,1} depends on both. Context functions ground facts in observable evidence, essential for manipulation immunity (Sec. 3.3). We will clarify in the revision.
>
> > Q3: Why does accuracy decrease with larger agent models?
>
> The judge is fixed; larger agents are stronger adversaries. From 7B→72B, LLM judges degrade from 78-85% to 62-71%. FormalJudge remains robust at 91-96% (+10-26 % over best baseline), since verification uses grounded facts immune to persuasion.
>
> > Q4: Can FormalJudge defend against multi-turn jailbreak attacks?
>
> Yes. Our existing benchmarks already cover multi-turn attacks, and we provide two new experiments.
>
> - **Existing multi-turn coverage.** VitaBench Robustness includes L3 (context exploitation) and L4 (multi-round history attacks) subsets.
>
> | Level | FormalJudge (avg) | Best Baseline | Δ |
> |-------|-------------------|---------------|---|
> | L3 context exploit. | 89.0% | 71.5% | +17.5 |
> | L4 multi-round hist. | 89.5% | 72.8% | +16.7 |
>
>
> On AgentSafetyBench multi-turn subset, Dafny achieves highest alignment on 6/7 agents (avg 75.3% vs. second-best 68.7%).
>
> - **X-Teaming (SOTA dynamic multi-turn jailbreak).** X-Teaming uses collaborative attacker agents for adaptive attacks. We evaluate on **n=72** scenarios per condition:
>
> | Agent | Dafny | Python | NL | Direct Judge | CoT |
> |---|---|---|---|---|---|
> | Claude-Sonnet-4.5 | **98.6%** | 93.3% | 85.8% | 88.6% | 73.5% |
> | GPT-4o | **88.9%** | 72.2% | 86.1% | 79.7% | 59.7% |
>
> - **OpenClaw compositional safety.** On OpenClaw (310K+ GitHub stars), we construct 42 multi-turn attack scenarios (privilege escalation, prompt injection, data exfiltration, multi-agent coordination) with a weaker (GPT-4o) and stronger (Claude-Opus-4-6) judge:
>
> | Method | GPT-4o (judge) | Claude-Opus-4-6 (judge) |
> |---|---|---|
> | FormalJudge | **90.5%** | **88.1%** |
> | LLM Judge | 61.9% | 66.7% |
>
> FormalJudge outperforms LLM Judge by +22-29 % regardless of judge capability, **confirming the advantage stems from the formal verification architecture rather than model strength.** Full experimental details and reproduction scripts will be included in our paper.
>
> We have tried our best to address every concern with new experiments and data. **If these results have helped resolve your reservations, we truly hope to receive even greater support from you.**
>
> We are happy to clarify any remaining questions!

---

> > ### Author Rebuttal · Reviewer_H2wa · 2026-04-04
> >
> > Thank you for the rebuttal and my concerns have been addressed. Please enclose the rebuttal results in the updated version.

---

> > > ### Author Response · Authors · 2026-04-06
> > >
> > > Thank you for confirming that your concerns have been addressed! **We will enclose all rebuttal results in the updated version as requested**, including the voting experiments, X-Teaming multi-turn jailbreak results, and OpenClaw agent safety evaluations.
> > >
> > > We appreciate the effort you invested in raising these questions; **the new features they motivated substantially strengthened the paper** (particularly the multi-turn defense and voting analysis).

---

### Decision · Program_Chairs · 2026-04-30

**Decision:**

Accept (regular)

**Comment:**

This paper proposes Formal-of-Thought for LLM oversight, a framework
where instead of using LLM judges directly, they instead have LLM
decompose the inputs and then use Dafny and Z3 for a formal verification
process. Reviewers agreed that the motivation is strong: there are inherent
problems with using LLM judges, and the method of using formalization is
novel and can address this. Also, the results are comprehensive and
include valuable additional experiments regarding weak to strong
generalization and iterative refinement. Reviewers acknowledged that
LLMs are used in the process, so there are inherent limitations with how
well the decomposition of the natural language into specification is a
difficult problem. Some parts of the framework can also be more clearly
specified. The rebuttals addressed the author concerns and were well-
received by all reviewers who had a positive assessment of the paper. So,
the decision is to accept.